# On the Photo-Eradication of Methicillin-Resistant *Staphylococcus aureus* Biofilm Using Methylene Blue

**DOI:** 10.3390/ijms24010791

**Published:** 2023-01-02

**Authors:** Irena Maliszewska, Anna Zdubek

**Affiliations:** Department of Organic and Medicinal Chemistry, Faculty of Chemistry, Wrocław University of Science and Technology, Wybrzeże Wyspiańskiego 27, 50-370 Wrocław, Poland

**Keywords:** aPDT, biofilm, methylene blue, gold nanoparticles, biofilm washing, disinfectant

## Abstract

This work compared the effectiveness of several Methylene Blue (MB)-based protocols for photo-eradication of biofilms formed on the surface of the glass and stainless steel discs by *S. aureus* MRSA isolates using a diode laser (λ = 665 nm; output power 40 mW; energy fluence was 189 J cm^−2^). The results obtained showed that MB alone, up to a concentration of 62.5 mgL^−1^, had limited photo-bactericidal activity. It was possible to enhance the activity of MB using two types of spherical gold nanoparticles of similar sizes, 15 ± 3 nm/20 ± 3 nm, but differing in the method of their synthesis and stabilization. The enhancement of the photodestruction effect was related to the increased production of hydroxyl radicals by the MB+gold nanoparticles mixture, and this mixture showed dark cytotoxicity against the *cocci* studied. Effective destruction (mortality above 99.9%) of the biofilms formed by MRSA isolates was also possible without the use of gold nanoparticles, but the concentration of MB had to be at least 125 mgL^−1^. A highly efficient protocol of photodestruction of biofilms, consisting of triple exposure of biofilms to laser light in the presence of MB alone, combined with the removal of dead bacteria protecting deep layers of pathogens against photosensitization, was also described.

## 1. Introduction

Our understanding of the growth of bacteria in the environment has changed over the years. It is now known that biofilm formation is the preferred lifestyle for most bacterial species, and approximately 40–80% of the bacterial cells on Earth can form these communities [1]. Biofilm could be defined as a structured community of microbes that adhere to each other and/or to the abiotic surface, and that is often embedded in a self-produced matrix of extracellular polymeric material [2]. It is believed that biofilms develop on all surfaces when in contact with nonsterile water and appear to be self-sustaining once formed. Due to serious medical and economic risks, biofilm formation is a problem in different sectors, ranging from public health to the food industry [1]. Stainless steel device surfaces (furniture, sensor probes, washing sinks, hand washing sinks, waste trolleys, instruments, etc.) are supposed to be the main sites for microbial adhesion and biofilm formation [3]. It is also common for a bacterial biofilm to form on glass surfaces since glass products are used in many areas of human life. On a daily basis, we can have contact with glass table tops, glass cabinets and walls, and glass displays (e.g., mobile phones).

The structure of the biofilm allows bacteria to resist many types of environmental stress, including UV radiation, lack of nutrients, and the presence of antimicrobial agents [4]. Previously, it was found that biofilm bacteria were 10–1000 times more resistant to chemical treatments than planktonic forms [5,6,7].

A simple strategy to effectively remove biofilms from a variety of surfaces may appear to be to increase the concentration of the antimicrobial agent significantly. It was previously shown that this approach was not only costly but also environmentally unfriendly and did not guarantee the elimination of pathogens [8]. Therefore, the elimination of biofilms can be achieved through the combined use of treatments characterized by different mechanisms of action. For example, in the food industry, cleaning pipelines and production equipment is performed using a cleaning-in-place (CIP) technique. This method is based on the sequential use of sodium hydroxide and nitric acid and combining this protocol with disinfectants.

Thus, removing biofilms from a variety of surfaces remains difficult and can be costly. Consequently, the development of new biofilm elimination protocols (Figure 1) has become the main area of research in this field [9,10,11,12]. One technique that appears to be a promising procedure to control biofilm growth effectively is antimicrobial photodynamic inactivation (aPDI), which is based on the production of reactive oxygen species using a light-activated photosensitizer (PS). It is considered that the antibiofilm activity involves two steps [13]. In the first step, the photosensitizer binds to the biofilm matrix and can be fully sequestered by EPS, while in other biofilms, PS may partially pass through the EPS and contact bacterial cells. Some photosensitizers bind to the cell surface, while other types of PS can pass through the cytoplasmic membrane and get into the cytoplasm. After irradiation of the photosensitizer, regardless of its location, ROS are generated to initiate the second step, i.e., multitarget oxidative damage. ROS attacks a variety of molecules, including targets in the biofilm matrix (e.g., polysaccharides), on the cell surface (e.g., lipids), and inside cells (e.g., proteins and DNA), destroying the structure of the biofilm matrix and killing pathogenic cells [14]. It should be emphasized that many cellular components (proteins, fatty acids, DNA, etc.) are the target of aPDT-induced oxidative stress after light activation of the photosensitizer; hence, the lack of sensitivity of microorganisms to aPDT (as it is observed in the case of bacterial resistance to antibiotics) is unlikely.

This study, for the first time, compared the effectiveness of several Methylene Blue (MB) photo-bactericidal protocols against biofilms formed on the surface of the glass and stainless steel discs by *S. aureus* MRSA isolates. The first set of methods was based on changing the dye concentration and synergistic effect of MB and gold nanoparticles, while the second protocol was a multistep procedure that involved not only photosensitizing the bacteria but also washing biofilms (removal of dead cells). The possible mechanism of enhancing the photocidal activity of MB by the gold nanoparticles has also been discussed.

## 2. Results

The antimicrobial susceptibility to antibiotics of the clinical isolates of *S. aureus* is shown in Appendix A. As can be seen in this table, the tested strains were resistant to penicillin (methcillin), but these bacteria were sensitive to gentamicin, erythromycin, and tetracycline.

The assessment of the susceptibility of biofilms formed by the studied isolates of *S. aureus* on the surface of the glass and stainless steel discs to photosensitization with MB was preceded by the determination of nontoxic concentrations of this photosensitizer and two types of gold nanoparticles designed as AuBNPs and AuChNPs (dark toxicity). The nontoxic concentration has been defined as the highest concentration of the MB, AuBNPs, and AuChNPs that, under the experimental conditions, caused a decrease in cell viability by no more than 20% (see Appendix A). As can be seen in these tables, in most cases, MB at a concentration of 31.25 mgL^−1^ was found to be nontoxic to the tested *coccus*. It was estimated that AuBNPs and AuChNPs were nontoxic, most often at a concentration of 10 ppm.

The effectiveness of photoeradication of biofilms developed on the surface of the glass and stainless discs after 30 min of laser light irradiation (energy fluence = 189 Jcm^−2^) was shown in Figure 2a–d and Figure 3a–d. As can be seen in Figure 2a–d, MB alone at a concentration of 31.25 mg L^−1^ was ineffective as a photosensitizer against the *S. aureus* MRSA isolates.

The most effective photocidal activity of this dye against biofilms formed on the glass within 24 h, resulting in a reduction in the number of viable *S. aureus* 3374 by no more than 0.19 ± 0.02 log_10_ units (bacterial mortality was approximately 67%). A similarly ineffective biocidal activity of MB was found for biofilms formed on the glass surface within 48 h with the highest reduction in the number of viable bacteria of 0.2 ± 0.02% log_10_ units (bacterial mortality was not higher than 63%) (Figure 2a). Figure 2a–d showed that the addition of gold nanoparticles to the MB increased the elimination of the *S. aureus* biofilm, and this phenomenon depended on the MRSA isolate and the time of biofilm formation (age of biofilm). It should be noted that the results of biofilm destruction efficiency were still insufficient since the highest reduction in cfu of 1.21 ± 0.02 log_10_ units was observed for 24-h biofilm formed by *S. aureus* 3515 using MB+AuChNPs as photosensitizer (bacterial mortality rate was about 94 ± 1%). Under the same experimental conditions, the viability of biofilms formed on the surface of glass discs for 48 h decreased by 0.51 ± 0.03 log_10_ units, i.e., cell mortality reduced to 69 ± 2%.

The cell killing efficiency (regardless of the age of the biofilm formed on stainless steel) in the presence of MB alone was also insignificant and, in most cases, did not exceed 40% (Figure 3a–d). The only exception was the biofilms formed by isolate 3515 after 24 and 48 h of incubation, as bacterial mortality was 68 ± 2 (cfu reduction of 0.49 ± 0.03 log_10_ unit) and 50 ± 2% (cfu reduction of 0.38 ± 0.03 log_10_ unit), respectively (Figure 3c).

The low photocidal efficacy of aPDT described above inspired an increase in the MB concentration as a photosensitizer to 62.5 mg L^−1^. To facilitate analysis of the obtained results, the degree of death of the isolates was expressed as the average reduction in the number of viable cells (%), which was calculated as the arithmetic means of the mortality level of the strains marked 3374, 3375, 3515, 3690 (Appendix A). The results obtained showed that the average killing effect of the studied *S. aureus* isolates was significantly higher compared to MB at a concentration of 31.25 mg L^−1^.

The cell death rate caused by MB, MB + AuBNPs, and MB+AuChNPs averaged 88 ± 2, 99.5 ± 1.5, and 99.951 ± 0.005% for 24-h biofilms formed by the studied strains on the glass discs. LIVE/DEAD™ *Bac*Light™ Bacterial Viability Kit (Thermo Fisher Scientific, USA) for monitoring bacterial cell viability as a function of cell membrane integrity was used to evaluate the effectiveness of aPDI against biofilms formed on the glass surface during 24 h of incubation by *S. aureus* 3515 with MB, MB + AuBNPs and MB+AuChNPs as photosensitizers. Dead *cocci* (with damaged cell membranes) were stained red (Figure 4a–d). The increasing amount of red color in the microscopy images corresponded to a higher efficiency of biofilm destruction.

For biofilms growing on a glass surface for 48 h, the photocidal activity of MB, MB+AuBNPs, and MB+AuChNPs averaged 81 ± 1.5, 90.8 ± 1.3, and 97.98 ± 0.95%, respectively (see Appendix A). Photoeradication of biofilms formed by the MRSA isolates on steel discs in the presence of MB, MB+AuBNPs, and MB+AuChNPs as photosensitizers occurred at a similar level (compared to biofilms developed on the glass surface) and the highest level of mortality was observed, reaching 99.948 ± 0.021%, when MB+AuChNPs were used as a photosensitizing mixture (Appendix A).

Another photocidal protocol was performed using MB as photosensitizer at a concentration of 125 mg L^−1^, and it was found that this procedure resulted in extremely high efficiency in destroying pathogens. Regardless of strain, type of material surface, or the age of the biofilm, the mortality rate was higher than 99.9% after exposure of the MRSA isolates to the light dose of 189 J cm^−2^ (Appendix A). The presence of gold nanoparticles in the photosensitizing mixture resulted in a lethal effect (the number of bacteria was below the detection level). 

Subsequent photoelimination of the bacteria consisted of three exposures of the biofilms to laser light in the presence of MB alone at a nontoxic concentration of 31.25 mg L^−1^, but in combination with the removal of dead cells (see: *Effect of photodynamic therapy on the viability of the S. aureus biofilm*; Procedure 2). The effectiveness of photodestruction of biofilm in these experimental conditions depended on the isolate, the age of the biofilm, and the type of surface. When biofilms that grow on the glass surface were exposed to laser light, the average reduction in the number of viable cells averaged 99.965 ± 0.03 and 99.94 ± 0.03% for the 24 and 48 h structures, respectively (Appendix A). Confirmation of the high efficiency of this method is shown in Figure 5a–c (the bacteria with damaged membranes stain fluorescent red). A more effective pathogen photodestruction was achieved on steel surfaces, and bacterial mortality averaged 99.979 ± 0.030 and 99.934 ± 0.032% for 24- and 48-h biofilms, respectively (Appendix A). The addition of gold nanoparticles to the photosensitizing mixture resulted in a lethal effect (the number of bacteria was below the detection level). 

In the last set of our experiments, the effectiveness of removing biofilms formed on glass/steel surfaces by MRSA isolates was examined using two commercially available disinfectants (designed as ① and ②). The results obtained are collected in Appendix A. As can be seen in this table, the cell death rate caused by disinfectant ① averaged 80 ± 2 and 69 ± 3% for 24- and 48-h biofilms formed by the isolates in glass discs, respectively. When biofilms developed on the surfaces of steel discs within 24 and 48 h, the biocidal efficacy of the product designed as ① was 46 ± 3 and 38 ± 4%, respectively. Under the same experimental conditions, the mortality of 24- and 48-h biofilms formed by the isolates on a glass disc using disinfectant ② was, on average, 92 ± 2 and 87 ± 3%, respectively. The product designed as ② removed an average of 68 ± 3 and 62 ± 3% of viable cells from the 24 and 48 h biofilms developed on the steel surface, respectively.

## 3. Discussion

This paper compared the effectiveness of several photobactericidal protocols against biofilms formed on glass and stainless steel discs by four MRSA strains of *S. aureus.* Methylene blue was applied as a photosensitizer due to its many advantages, such as good solubility in water, a wide spectrum of light absorption, high efficiency of ROS release, and affordable price [15,16]. Stainless steel and glass surfaces were chosen for biofilm formation as both types of materials are found in many sectors of public life.

Initially, photoeradication of biofilms formed by the MRSA strains was carried out on the surfaces of glass and stainless steel discs in the presence of this dye at concentrations of 31.25 and 62.5 mg L^−1^. In general, the effectiveness of these photoeradication protocols, regardless of the pathogenic isolate and the type of disc surface, was insignificant. It should be noted that our results did not support the data reported previously, indicating an extremely effective photodestruction of *S. aureus* biofilm in the presence of MB at relatively low concentrations. Tanaka et al. [17] demonstrated that the optimal concentration of MB as a photosensitizer against MRSA strains was 0.1 mM (that is 31.98 mg L^−1^). The studies by Usacheva et al. [18] showed that MB concentrations of MB between 0.025 and 0.044 mM (that is, 8–14 mg L^−1^) were the most effective in combating the studied *cocus*. Pérez-Laguna et al. [19] reported that irradiation of the *S. aureus* biofilm with a light-emitting diode (625 nm; 18 J cm^−2^) in the presence of MB at a concentration of 64 mg L^−1^ resulted in a reduction in cell number by 6 log_10_ in colony-forming units. On the other hand, Biel et al. [20] presented the results of the combating of the bacterial biofilm of *S. aureus* from the lumen of the endotracheal tube. These authors used MB at a concentration of 500 mg L^−1^ and non-thermal activated light (664 nm; fluidity coefficient 150 mW cm^−1^). After a single light treatment, the bacterial count has been shown to decrease by at least 99.9% (a 3 log_10_ reduction in colony-forming units). It is worth noting that MB alone (without light activation) reduced the number of bacteria by approximately 90% (a reduction of 1 log_10_ in colony-forming units).

It is not possible to explain these contradictory results relating to the MB concentration that causes a satisfactory level of biofilm photomortality. It seems that it would be necessary to standardize the procedure of photoeradication of bacterial biofilms.

To enhance the efficiency of photodestruction of biofilms formed by *S. aureus* isolates, two types of spherical gold nanoparticles of similar sizes, 15 ± 3 nm/20 ± 3 nm, but differing in the type of stabilizer were used. The biogenic gold nanoparticles were synthesized by the cell-free filtrate from *Trichoderma koningii*. This mycological filtrate played an essential role not only in the reduction of Au(III) ions but also in the formation and stabilization of gold nanoparticles [21]. The second type of gold nanoparticles were polyethyleneimine-stabilized structures.

The results obtained clearly showed that the presence of gold nanoparticles in the photosensitizing mixture improved the inactivation of the biofilms developed on the glass and steel discs, and this effect was highly strain-dependent. Furthermore, young biofilms were more sensitive to photodestruction compared to older structures, and these observations are consistent with the results presented by Chen et al. [22]. Furthermore, it seems that more intense biofilm growth was on the surface of glass discs, and these cultures were more sensitive to laser light photosensitization. It has already been suggested that the efficiency of biofilm formation (as well as its removal) may be related to the surface nature [22,23]. Cell adhesion to the surface is one of the main factors that influence the formation of biofilms in various materials [24,25,26].

The use of MB as a photosensitizer at a concentration of 62.5 mg L^−1^ has been shown to allow the protocol to be accepted as compliant with the requirements of the American Microbiological Society (AMS), but only for young biofilms in the presence of AuChNPs in the photosensitizer mixture (at least 99.9% cell death is required for the procedure to be considered antibacterial [22,27]) (see Appendix A). Chemically synthesized gold nanoparticles appeared to be a slightly better adjuvant.

Due to the unsatisfactory results of biofilm removal described above, in the next part of our experiments, aPDT was carried out using MB at a concentration of 125 mg L^−1^. It should be emphasized that the extremely high photoeradication efficiency of the MRSA pathogens (without the addition of gold nanoparticles) was observed, and this protocol met the requirements of the AMS. The presence of gold nanostructures in the photosensitizing mixture results in a lethal effect (the number of bacteria was below the detection level).

In the case of biogenic AuBNPs, a possible mechanism to increase the photobactericidal activity of MB may be the rapid photobleaching of this dye [28]. The effect of AuChNPs on MB photobleaching was also studied, but the obtained results did not clearly confirm this phenomenon, therefore, it was hypothesized that the interaction of MB and gold nanoparticles during photodynamic cell destruction may result in the more effective production of toxic oxidative species. It is known that in photosensitization, MB may have two different mechanisms of action [29]. The excitation of MB by light can induce the singlet and triplet stages of the molecule and transfer energy through electrons (type I mechanism) or energy (type II mechanism). The Type I mechanism can lead to the formation of superoxide anions (O_2_^•−^), hydroxyl radicals (HO^•^), and hydrogen peroxide (H_2_O_2_). The Type II mechanism proceeds through singlet oxygen, resulting mainly in the breakage of nucleic acids, mostly at the guanosine site. Most authors suggested that ^1^O_2_ is the major destructive factor in PDT in vivo (this anion is a short-lived radical) [30]. However, it should be emphasized that both type I and type II mechanisms may occur simultaneously, leading to comparable oxidative cell damage, and the ratio of these pathways depends on the cellular microenvironment in which MB accumulates, including the concentration of molecular oxygen [14,31]. Moreover, there may be differences in the mode of action of aPDI in biofilms compared to bacteria cultivated in planktonic cultures due to extracellular polymeric substances and a potentially reduced penetration depth throughout the biofilms [14].

Detection of harmful ROS and singlet oxygen that could be generated by MB alone and the mixture of MB+gold nanoparticles was performed using common methods (see Section 4.7). There was no significant increase in the production of superoxide anions, hydrogen peroxide, and singlet oxygen in the presence of MB and gold nanoparticles in comparison to MB alone.

Statistically significant differences in the level of hydroxyl radicals were only observed, but the obtained results were not unambiguous and depended on the MRSA isolate. For this reason, a fluorimetric assay with dichlorodihydrofluorescein diacetate was used to directly measure the redox state of the cell [32]. As can be seen in Figure 6, ROS production by MB remains at a similar level throughout the experiment. In the case of the mixture of MB and AuBNPs, the efficiency of oxidative species production increased compared to the control (the control was the bacterial suspension) by 29 ± 3, 32 ± 3, 35 ± 3, and 41 ± 3%, (using light doses of 31.5, 63, 126, and 189 J cm^−2^), respectively. Under the same experimental conditions, the mixture of MB and AuChNPs increased the formation of oxidative species by 34 ± 4%, 41 ± 4%, 46 ± 4.0%, and 50 ± 4% compared to the control. The obtained results clearly showed that the higher efficiency of aPDT in the presence of gold nanoparticles is the result of increased production of oxidizing compounds (probably hydroxyl radicals).

The dark toxicity of MB+AuBNPs and MB+AuChNPs was also studied, and it was found that these mixtures containing nontoxic concentrations of MB and gold nanoparticles showed bactericidal activity.

Finally, we proposed a simple photocidal procedure that involved triple exposure of the biofilm to laser light in the presence of a nontoxic concentration of MB alone (31.25 mg L^−1^), combined with the removal of dead bacteria that protected the deep cell layers from photosensitization. This method increased the efficiency of biofilm destruction compared to a single photosensitization in the presence of a nontoxic concentration of MB and a light dose of 189 J cm^−2^, and this efficiency met the requirements of the ASM. It should be noted that washing the biofilm removed both dead and viable bacterial cells. The number of viable cells that were present in the biofilm washing water did not exceed 3 ± 1 and 5 ± 1% of the total number of viable biofilm-forming pathogens on the glass and stainless steel discs, respectively. The protocol of photodestruction of biofilms formed on glass and steel surfaces by *S. aureus* MRSA isolates in the presence of MB at a concentration of 125 mg L^−1^, and the procedure that involved triple exposure of the biofilm to laser light has been proven to be effective disinfection methods. These simple and inexpensive techniques can be used to disinfect a variety of small surfaces (from medical areas, fitness salons, spas, and beauty salons, to public places). It is worth noting that these procedures were more effective compared to the efficiency of removing biofilms from glass/steel discs using commercially available disinfectants (the product designed as ① was a mixture of ethyl alcohol and propan-1-ol; the product designed as ② was quaternary ammonium salts).

The attractiveness of the methods of biofilm destruction proposed above may be further enhanced by the use of inexpensive LED technologies with the appropriate light length, which is readily available on the commercial market. Methylene blue has been used as a photosensitizing agent since the 1920s, and its administration has been approved by the U.S. Food and Drug Administration.

Our understanding of bacterial biofilm formation has advanced significantly over the past 20 years, leading to the development of several (in addition to aPDI) promising biofilm destruction strategies. Conventional strategies (mentioned in the Introduction chapter) to control biofilm growth most often include cleaning and disinfection strategies such as physical (clean-in-place) and chemical (using sodium hypochlorite, hydrogen peroxide, ozone, and peracetic acid) methods. It has been shown that mechanical treatment (cleaning in place) cannot remove all bacterial cells, while chemical treatment only partially eliminates bacterial cells and inhibits biofilm formation [33,34]. Therefore, physical-mechanical methods, such as surface scrubbing and the use of antimicrobials, must be combined to address this problem.

To effectively combat infections resulting from biofilms formation, various methods have been developed covering different aspects, such as: (i) changing the properties of susceptible surfaces to prevent biofilm formation; (ii) regulating signaling pathways to inhibit biofilm formation, and (iii) applying external forces to eradicate the biofilm. An excellent discussion of the applicability and limitations of these methods can be found in recent reviews [35,36,37].

Regarding changing surface properties that prevented bacterial contamination of the surface, Uneputty et al. [38] presented four categories of promising methods: anti-adhesive, contact active, biocide attached/biocide release, and topological modification. Particular attention was paid to antibiotic-free antibacterial strategies.

It is well known that biofilm formation is regulated by several signaling pathways, such as quorum sensing (QS) and nucleotide second messenger systems. Therefore, biofilm inhibitors can be designed by targeting relevant proteins in the respective regulatory systems to block their signaling pathways and thereby inhibit biofilm generation. To date, quite a few QS signaling molecules have been described, including autoinducer peptide (AIP), N-acyl homoserine lactones (AHLs), autoinducer 2 (AI-2), Pseudomonas quinolone signal (PQS), diffusible signal factor (DSF) [39]. Among them, AIP, AHLs, and AI-2 are arguably the most widely studied. The regulation of biofilm formation by c-di-GMP, c-di-AMP, and (p)ppGpp, and the corresponding inhibitors that have been developed to control the respective regulatory processes, was also studied against biofilm formation.

Several different physical and biochemical approaches as external pressures to eradicate biofilm formation were proposed. For example, physical methods, such as treatment with ultrasound and magnetic fields, can be applied to eradicate biofilms effectively. These external physical factors are simple and crude, but they can only eliminate biofilm to a certain extent. These methods are suitable for decontamination in the food industry, biofilm control in dentistry, and removal of biofilms adsorbed on tissue implants and medical devices, but cannot be used for infected tissues that require more gentle methods such as the use of phage lysines, degrading enzymes, or plant metabolites [40,41].

Although there are many strategies to control harmful biofilms, due to the complex composition of this microbial structure and its high resistance to stress, it is difficult to remove biofilm with a single method completely. Therefore, using a “mixed approach” by combining different strategies may be more effective in eradicating biofilms from different surfaces. We believe that understanding the details of the process of biofilm formation and regulation of its signaling pathways will enable the discovery of new targets that can be used to develop highly effective protocols, and future biofilm control research must be multidisciplinary.

## 4. Materials and Methods

### 4.1. Reagents

All chemicals were obtained from Sigma-Aldrich (Poznań, Poland) and Avantor Performance Materials Poland S.A. (Gliwice, Poland). Methylene blue (MB) was dissolved in deionized water at a final concentration of 1 g L^−1^. The reagent was sterilized by filtration (0.22-µm pore diameter membranes). A solution of 2′,7′-dichlorodihydrofluorescein diacetate in ethanol at a concentration of 2 mM was used to detect intracellular reactive oxygen. This solution was stored at −20 °C.

### 4.2. Gold Nanoparticles

Biogenic gold nanoparticles (AuBNPs) were biosynthesized using the cell-free filtrate obtained from *Trichoderma koningii* according to the procedure described by Maliszewska et al. [21]. Chemically synthesized AuChNPs were synthesized using ascorbic acid as a reducing agent in the presence of polyethyleneimine as a stabilizing agent.

### 4.3. Light Sources

In these studies, a semiconductor single-mode red diode laser was used, which provides an emission wavelength of 665 nm and standard light output of 40 mW (the light intensity of 105 mW cm^−2^).

### 4.4. Microorganisms and Biofilm Formation

Four isolates of *Staphylococcus aureus* designed as *S. aureus* 3374, *S. aureus* 3375, *S. aureus* 3515, and *S. aureus* 3690 were used as tested microorganisms. One colony of *S*. *aureus* was inoculated in 2 mL of Mueller–Hinton liquid medium. Thus, the suspension was incubated at 37 °C in the dark for 18–24 h. After that time, the suspension was centrifuged (5 min/6000 rpm). The supernatant was discarded, the bacterial pellet was suspended in sterile deionized water, and this suspension was adjusted to a McFarland 0.5 standard containing a bacterial concentration of 1.2–1.5 × 10^8^ CFU mL^−1^ (OD_550_ = 0.09–0.1). The standardized suspension of *S*. *aureus* was applied to the surface of glass or stainless steel discs (10 mm × 10 mm) and pre-incubated for 1 h in the dark at 37 °C. The discs were washed with sterile deionized water to remove non-adhesive cells. The discs were then flooded with 0.5 mL of Mueller–Hinton broth and incubated for 24 or 48 h in the dark at 37 °C (the age of these biofilms was established to be 24 or 48 h, respectively).

### 4.5. Effect of Photodynamic Therapy on the Viability of the S. aureus Biofilm

The evaluation of the effect of photodynamic inactivation on the viability of *S. aureus* biofilm cells was based on two procedures that are detailed below.

Procedure 1. After biofilm formation on the surface of glass/steel discs (see the protocol described above), MB, MB+AuBNsP, or MB+AuChNPs were dropped on the surface of each disc, and these samples were incubated in the dark at 37 °C for 30 min. Then all discs were exposed to diode laser lamp light (the peak power wavelength λ = 665 nm; output power 40 mW; power density of 105 mW cm^−2^) for 30 min. This study was carried out with 4 experimental groups: laser light alone (L+, n = 3); control probe; biofilm without photosensitizer), MB and laser light (L+MB, n = 3), MB+gold nanoparticles and laser light (L+MB+AuBNPs, n = 3), (L+MB+AuChNPs, n = 3). This procedure was performed for MB at concentrations of 31.25, 62.5, and 125 mg L^−1^. The viability of cells was estimated using a colony counting assay according to the method described in Appendix A (The dark cytotoxicity of Methylene Blue and gold nanoparticles).

Procedure 2. The inactivation of the *S. aureus* biofilm formed on the surface of the discs by aPDT was carried out according to the procedure described above with some modifications. In the first step, the biofilms (after dark incubation with MB at a concentration of 31.25 mg L^−2^) were treated with laser light for 10 min (energy fluence 63 J cm^−2^). After that, the biofilms were washed with sterile deionized water to remove the damaged cells from the surface of the biofilm, and fresh photosensitizer (MB) was added. All discs were again exposed to diode laser lamp light for 10 min (energy fluence of 63 J cm^−2^). This procedure was repeated three times according to the following scheme: 1. irradiation for 10 min/biofilm washing+adding a fresh MB; 2. irradiation for 10 min/biofilm washing+adding a fresh MB and 3. biofilm irradiation for 10 min/biofilm washing and estimate of bacteria viability. The viability of cells was estimated using a colony counting assay according to the method described in Appendix A (The dark cytotoxicity of Methylene Blue and gold nanoparticles). Deionized water, which was used to wash the biofilm culture, was seeded on an agar medium, and the number of viable cells was determined by a serial dilution method and administered as CFU mL^−2^.

### 4.6. Determination of Cell Integrity

The LIVE/DEAD^TM^ *Bac*Light^TM^ Bacterial Viability Kit for fluorescence microscopy (ThermoFisher Scientific, Rochester, NY, USA) was used to test the viability of bacteria. Visualization of the bacterial cells was performed under Olympus 60BX light microscope (Taunton, MA, USA) with appropriate excitation/emission filter cubes (excitation bandpass filter 460–490 nm, emission long pass filter >515 nm).

### 4.7. Measurement of Reactive Oxygen

Detection of reactive oxygen that could be generated by MB alone and the mixture of MB+gold nanoparticles was performed using the following protocols. The protocol based on the oxidation of luminol by superoxide anions, resulting in the formation of chemiluminescent light, was used to detect O_2_^•−^ (Superoxide anion detection kit; Sigma-Aldrich, Poznań, Poland). 2-[6-(4′-Hydroxy)phenoxy-3H-xanthen-3-on-9-yl]-benzoic acid was applied for the detection of hydroxyl radicals (according to the procedure described by Hwang et al. [42]. To detect hydrogen peroxide, 10-Acetyl-3,7-dihydroxyphenoxazine (Amplex™ Red Hydrogen Peroxide/Peroxidase Assay Kit; ThermoFisher Scientific, Rochester, NY, USA) was used.

### 4.8. The Effect of Gold Nanoparticles on the Singlet Oxygen Production

The effect of gold nanoparticles on the ability of MB to photoinduce singlet oxygen production was determined using a chemical probe 9,10-anthracenediyl-bis (methylene)dimalonic acid (ABMDMA) according to the procedure described by Zhao et al. [43]. The reaction was monitored spectrophotometrically by recording the decrease in optical density at 402 nm (λ_max_ of ABMDMA).

### 4.9. The Effect of Gold Nanoparticles on the Redox State of the Cell

Cell suspensions, after photoeradication using MB, MB+AuBNPs, and MB+AuChNPs, were incubated with a solution of 2’,7’-dichlorodihydrofluorescein diacetate (DCFDA) at a final concentration of 50 µM [44]. Then, ROS generation of ROSs was detected with the spectrofluorimetric method with excitation at 485 nm and emission at 520 nm using the SpectraMax Gemini spectrofluorimeter (Sunnyvale, CA, USA) and the SoftMax^®^Pro Software (Standard 5.x.; Molecular Devices, San Jose, CA, USA).

### 4.10. Statistical Analysis

Statistical analyzes have been performed using the STATISTICA data analysis software (version 13.3 PL; StatSoft, Kraków, Poland) and Excel. Quantitative variables were characterized by the arithmetic mean of the standard deviation or median or max/min (range) and 95% confidence interval. The statistical significance of the differences between the two groups was processed with Student’s t-test. In all the calculations, the *p*-value of 0.05 was used as the limit value.

## 5. Conclusions

We have compared several photoeradication protocols for biofilms formed by *S. aureus* MRSA isolates on the surfaces of glass and stainless steel discs. On the basis of the results obtained, the following conclusions were made: 1. the efficiency of biofilm formation and its photo-destruction depended on the bacterial isolate, surface nature, age of the biofilm, and MB concentration; 2. the presence of gold nanoparticles increased the photodestruction efficiency of biofilms, and this phenomenon was related to the increased production of hydroxyl radicals by mixtures of MB and gold nanoparticles; moreover these mixtures showed dark cytotoxicity against bacterial cells; 3. an effective photoelimination (more than 99.9%) of biofilms formed by MRSA isolates was possible, but the concentration of MB must be at least 125 mg L^−2^ or a biofilm destruction procedure involving triple photosensitization of microorganisms in the presence of MB at a non-toxic concentration of 31.25 mg L^−2^, in combination with the removal of dead pathogens, should be applied.

## Figures and Tables

**Figure 1 ijms-24-00791-f001:**
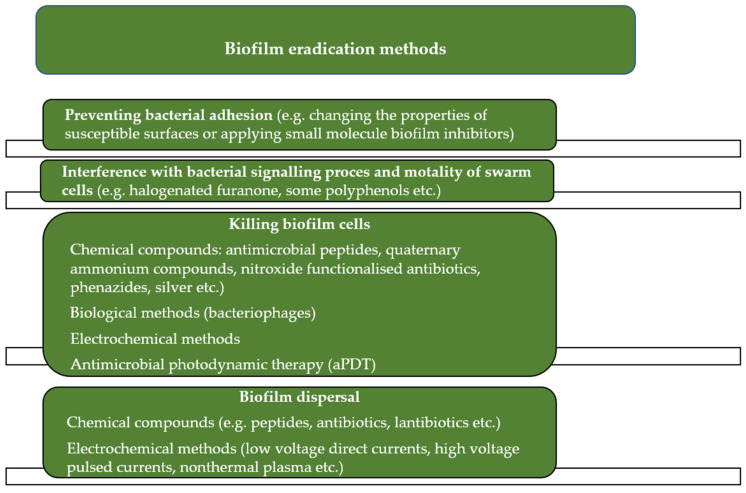
The strategies for controlling biofilm formation.

**Figure 2 ijms-24-00791-f002:**
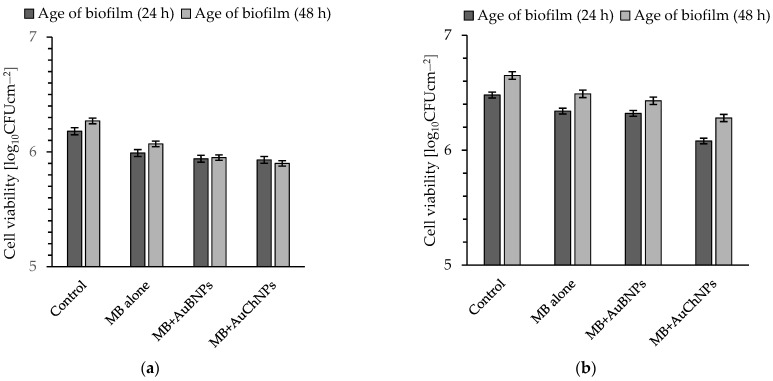
Effect of MB alone, MB + AuBNPs, and MB+AuChNPs on the viability of biofilm formed on the surface of glass discs by *S. aureus* 3374 (**a**); *S. aureus* 3375 (**b**); *S. aureus* 3515 (**c**); *S. aureus* 3690 (**d**) after exposure to laser light for 30 min (energy fluence was 189 J cm^−2^). Average ±SD of three independent experiments is shown; in all groups, *p* < 0.05. Age of biofilm: (24 h) = biofilm formed on the glass surface within 24 h; (48 h) = biofilm formed on the glass surface within 48 h.

**Figure 3 ijms-24-00791-f003:**
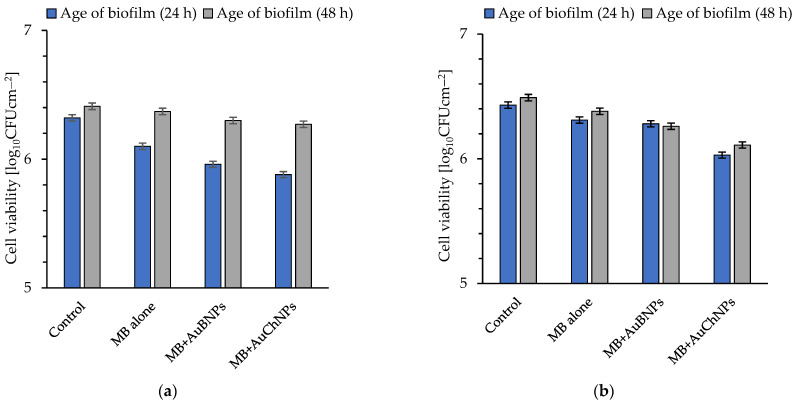
Effect of MB (Methylene Blue), MB + AuBNPs, and MB+AuChNPs (Methylene Blue + gold nanoparticles mixture) on the viability of biofilm formed on the surface of stainless steel discs by *S. aureus* 3374 (**a**); *S. aureus* 3375 (**b**); *S. aureus* 3515 (**c**); *S. aureus* 3690 (**d**) after exposure to laser light for 30 min (energy fluence was 189 J cm^−2^). Average ±SD of three independent experiments is shown; in all groups, *p* < 0.05. Age of biofilm: (24 h) = biofilm formed on the steel surface within 24 h; (48 h) = biofilm formed on the steel surface within 48 h.

**Figure 4 ijms-24-00791-f004:**
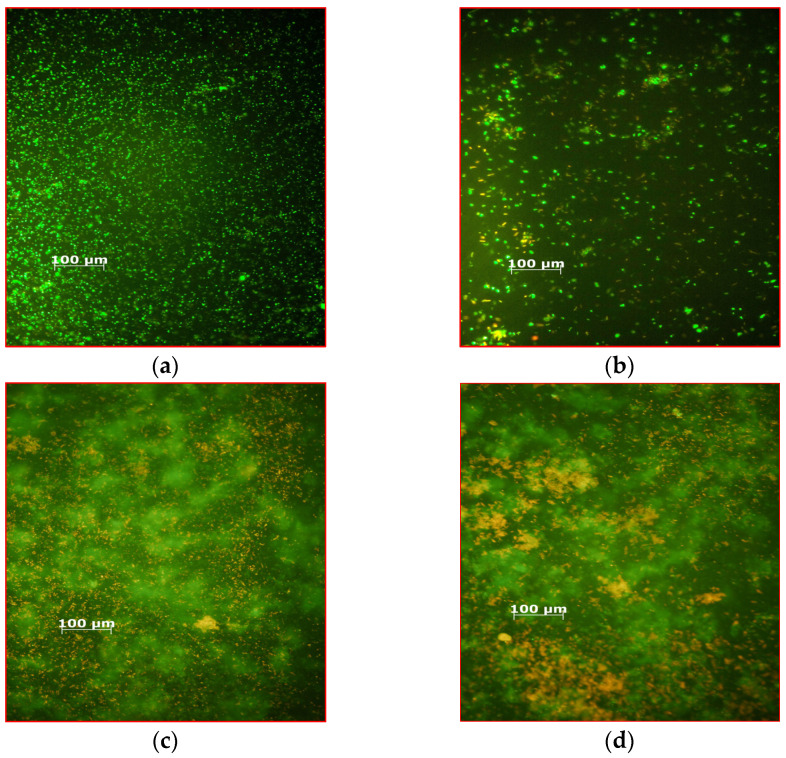
Fluorescence microscopy images of biofilm formed on the glass surface by *S. aureus* 3515 before aPDI (**a**); after photodynamic inactivation with MB alone as a photosensitizer (**b**); after photodynamic inactivation with MB + AuBNPs as a photosensitizer (**c**); after photodynamic inactivation with MB+AuChNPs as a photosensitizer (**d**). In the presence of the SYTO 9 and propidium iodide mixture, bacteria with intact cell membranes stain fluorescent green, whereas bacteria with damaged membranes stain fluorescent red. Scale bar: 100 μm.

**Figure 5 ijms-24-00791-f005:**

Fluorescence microscopy images of biofilm formed on the glass surface by *S. aureus* 3375 after the first (**a**); second (**b**) and third (**c**) exposures to laser light in the presence of MB alone at a nontoxic concentration of 31.25 mg L^−2^. In the presence of the SYTO 9 and propidium iodide mixture, bacteria with intact cell membranes stain fluorescent green, whereas bacteria with damaged membranes stain fluorescent red. Scale bar: 100 μm.

**Figure 6 ijms-24-00791-f006:**
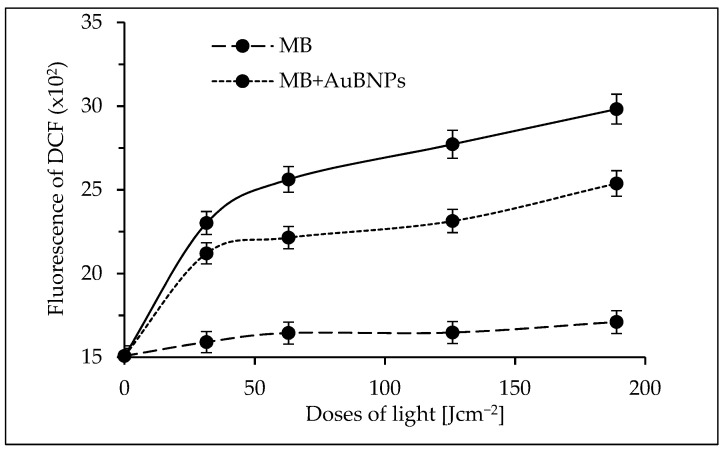
Effectiveness of generating oxidative species by MB alone, MB+AuBNPs, and MB+ChNPs as photosensitizers. The dichlorodihydrofluorescein diacetate as a fluorescent probe was used. Average ±SD of three independent experiments is shown.

## Data Availability

Data is contained within the article.

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
