# Peer review of "On the Photo-Eradication of Methicillin-Resistant Staphylococcus aureus Biofilm Using Methylene Blue"

_ijms, 2023, doi:10.3390/ijms24010791_

Round 1

Reviewer 1 Report

The authors compared the effectiveness of several MB photo-bactericidal protocols against biofilms formed on the surface of glass and stainless-steel discs by S. aureus MRSA isolates. This article has a reasonable structure and certain innovation. I recommend acceptance of this article after minor revisions.

1. Please add some photos, SEM images or visual effect diagram of the biofilm and mixture (MB + two types of spherical gold nanoparticles) during the experiments, not all data graphs.

2. As your said in conclusions part “the presence of gold nanoparticles increased the photodestruction efficiency of biofilms and this phenomenon was related to the increased production of hydroxyl radicals by mixtures of MB and gold nanoparticles” Could you please provide relevant evidence? Although you have measured that in the presence of gold nanoparticles, aPDT is more efficient as a result of increased production of oxidizing compounds, there are many other kinds of oxidizing compounds.

3. Whether the products of photodestruction have any other side effects.

Author Response

I thank you for the reviewing my manuscript and I am sending answers to all suggestions in the attached file.

Reviewer 2 Report

This paper reported the study of several Methylene Blue (MB)-based protocols for photo-eradication of biofilms formed by S. aureus MRSA isolates on the surfaces of glass and stainless-steel discs. Especially, the combination of MB with gold nanoparticles showed enhanced photo-destruction effect even in dark as the increased production of hydroxyl radicals. The topic fits the scope of the journal and will benefit the development of advanced biofilm-removing protocols. In general, the manuscript is well-organized, and the background references are sufficient to support the study design. Also, the experiments can support the conclusions. Even though, some issues are required to be addressed before its publication on International Journal of Molecular Sciences.

1. The current protocols of controlling biofilm growth are suggested to be introduced with a figure illustration in the introduction section.

2. The cytotoxicity of MB-based protocols on human normal cells are required to be assayed to evaluate the potential effects of these protocols on human health.

3. The development strategy of new biofilm destruction protocols are suggested to be discussed and prospected.

Author Response

i thank you for reviewing my manuscript and I am sending answers to all suggestions in the attached file.
